# Breastfed and Formula-Fed Infants: Need of a Different Complementary Feeding Model?

**DOI:** 10.3390/nu13113756

**Published:** 2021-10-24

**Authors:** Margherita Caroli, Andrea Vania, Maria Anna Tomaselli, Immacolata Scotese, Giovanna Tezza, Maria Carmen Verga, Giuseppe Di Mauro, Angelo Antignani, Andrea Miniello, Marcello Bergamini

**Affiliations:** 1Independent Researcher, Francavilla Fontana, 72021 Brindisi, Italy; 2Independent Researcher, 00162 Rome, Italy; 3Nutrition Unit, Department of Prevention, Azienda Sanitaria Locale Brindisi, 72100 Brindisi, Italy; annat76@libero.it; 4ASL Salerno, Campagna, 84022 Salerno, Italy; scotese.ped@libero.it; 5F. Tappeiner Hospital, Merano, 39012 Bolzano, Italy; giovanna.tezza@gmail.com; 6ASL Salerno, Vietri Sul Mare, 84019 Salerno, Italy; vergasa@virgilio.it; 7ASL Caserta, Aversa, 81031 Caserta, Italy; presidenza@sipps.it; 8Department of Food Science, University of Naples Federico II, 80100 Napoli, Italy; angelo.antig@gmail.com; 9School of Allergology and Immunology, University of Bari, 70124 Bari, Italy; miniello_andrea@yahoo.it; 10AUSL Ferrara, 44121 Ferrara, Italy; marcelloberga54@gmail.com

**Keywords:** weaning, complementary feeding, breastfeeding, formula feeding, cow milk, nutritional needs, infancy, toddlers

## Abstract

Suboptimal nutrient quality/quantity during complementary feeding (CF) can impact negatively on infants’ healthy growth, even with adequate energy intake. CF must supplement at best human milk (HM) or formulas, which show nutritional differences. Considering this, a differentiated CF is probably advisable to correctly satisfy the different nutritional needs. To assess whether current needs at 6–24 months of age can still be met by one single CF scheme or different schemes are needed for breastfed vs. formula/cow’s milk (CM) fed infants, protein, iron and calcium intakes were assessed from daily menus using the same type and amount of solid food, leaving same amounts of HM and follow-up formula at 9 and again 18 months of age, when unmodified CM was added. Depending on the child’s age, calcium- and iron-fortified cereals or common retail foods were used. The single feeding scheme keeps protein intake low but higher than recommended, in HM-fed children while in formula/CM-fed ones, it achieves much higher protein intakes. Iron Population Recommended Intake (PRI) and calcium Adequate Intakes (AI) are met at the two ages only when a formula is used; otherwise, calcium-fortified cereals are needed. ESPGHAN statements on the futility of proposing different CF schemes according to the milk type fed do not allow to fully meet the nutritional recommendations issued by major Agencies/Organizations/Societies for all children of these age groups.

## 1. Introduction

The complementary feeding (CF) period accounts for a difficult and vulnerable time, a time when limited gastric capacity is combined with high energy needs to ensure growth and health. Suboptimal nutrient quality and quantity can have a negative impact on growth and neurodevelopment, even when overall energy intake is adequate [1]. It is, therefore, necessary that complementary foods supplement as best as possible human milk or formulas.

Human breast milk (HM) is the ideal nutrition for infants, but some of them are formula fed and, although the composition of all baby formulas (either derived from cow milk [CM] or with vegetal protein source) has improved a lot over time, differences still remain between formulas and HM, not only in the amount of some macro and micronutrients, but also in terms of functional factors that are not fully understood yet [2].

Table 1 summarises the differences in main macro and micronutrients among HM, follow-up formulas, young child formulas (YCF), and CM.

Another key difference between formulas and HM is that formulas have a pre-set composition and a taste that is always the same (although it may differ from one formula to another one), whereas HM varies in composition and taste—not only throughout the whole breastfeeding period, but also during the day, and during each feeding—depending on a wide range of factors, including age of the infant, mother’s diet (at least in part), her age, weight, and, most likely, her genetic background [6]. Research on the macronutrient composition of HM was mainly carried out in the 1980s and 1990s [7,8,9] and was mainly focused on the composition of milk in the first 6 months of lactation, whereas there are far fewer studies on the composition of HM beyond 6 months up to (and beyond) the first year of life [10,11,12].

Studies on HM composition, conducted in different countries across five continents, show a fairly similar range of energy and nutrient values, from which standard values are often extrapolated to make calculations easier when assessing energy and nutrient intakes. In general, during the first year of life, protein content of HM decreases as breastfeeding goes on, while fat and carbohydrate contents remain stable [11,13,14]. A recent, interesting, study reports longitudinal changes in the macronutrient concentration of milk from healthy women from 0 to 48 months of lactation. The results of this study suggest that, after 18 months of breastfeeding, concentrations of lipids and proteins increase compared to that of milk produced in the first 12 months (lipids 5.80 g/100 mL at 24 months vs. 3.46 g/100 mL at 12 months; proteins 1.24 g/100 mL at 24 months vs. 1.00 g/100 mL at 12 months) while the concentration of carbohydrates decreases (6.6 g/100 mL at 24 months vs. 7.1 g/100 mL at 12 months). Then, from 24 months to 48 months, macronutrient concentrations remain stable [15]. However, the study has a sampling bias, since milk analysis was performed on one single sample per day, taken in the morning and, therefore, not representative of the average daily composition. Another study [16], biased by the inclusion of only 19 women, also reported that the protein content increased during the second year of lactation (1.6 g/100 mL at 11 months vs. 1.8 g/100 mL at 17 months). Given these methodological problems, the results of these studies, albeit very interesting and apparently in line with the increased growth needs of children, need to be corroborated before they can be considered conclusive and useful in clinical practice.

Differences between HM and formulas are also qualitative, e.g., in terms of casein and serum protein types, aminoacidic profile and fatty acids. Finally, HM contains many molecules and components with a range of biological functions that are absent and so far, non-replaceable, or anyway present in different amounts, in formulas [13].

All these differences must be taken into account when considering complementary foods, as they may have. different effects on body composition from the very first months of life into adulthood and may influence long term health outcomes.

Many years ago, it was convincingly demonstrated that formula-fed infants gain more weight (but not more length) in the first year of life when compared to breastfed infants [17,18]. What is new is that it has been shown quite recently [19] that the weight difference is due to the larger amount of lean mass compared to that of breastfed infants, which is observed from the age of 3 months and is still detectable at 7 months of age. The amount of fat mass is similar in the two groups, but differs in distribution, as breastfed infants have a greater amount of subcutaneous fat than formula-fed infants, who exhibit a greater amount of visceral fat [20], with the first condition (more subcutaneous than visceral fat) seeming to be a protective factor in the development of metabolic changes at later ages [21,22].

The reasons for this different body composition are still unclear. They could be linked, for instance, to the different macronutrient composition: a higher protein intake from infant formulas (particularly, a higher intake of casein, which contains the highest amount of insulinogenic amino acids) [23,24] could stimulate greater insulin and IGF-1 secretion and promote visceral fat deposition [25]. A totally different explanation might be the ascertained fact that formula-fed infants have a different profile of appetite-regulating hormones than breastfed infants: one study has shown that breastfed infants have lower serum levels of ghrelin, leptin and insulin, hormones associated with fat mass and its changes as compared with formula-fed infants [26]. In addition, the higher fat content in hind-milk vs. fore-milk in breastfed infants may also contribute to appetite regulation in addition to hormones [27].

Another important difference is that HM has only 13% casein, the lowest casein concentration of all the mammalian species studied, which could explain the slow growth of breastfed human infants [28].

Given the different body composition observed in the two different feeding patterns (breastfeeding vs. formula feeding) and given the different nutritional properties of HM and formulas, CF most likely needs to be differentiated between the two groups precisely because it is “complementary” to two very different foods. The European Society for Paediatric Gastroenterology, Hepatology and Nutrition (ESPGHAN) has acknowledged this difference and its implications on terms of CF but has stated that “Because the composition and health effects of breast milk differ from those of infant formula, on a theoretical basis it may seem sensible to give different recommendations on CF to breastfed versus formula-fed infants. Despite these theoretical considerations, devising and implementing separate recommendations for the introduction of solid foods for breast-fed infants and formula-fed infants may, however, present practical problems and cause confusion among caregivers” [29]. However, this statement is not based on any scientific evidence, nor have any studies been conducted since the publication of the ESPGHAN position paper to confirm these concerns, so it remains an arbitrary assumption.

## 2. Objectives

Aim of this paper is to assess whether, in light of the new knowledge, the nutritional needs and recommendations proposed by the major Agencies/International Health Organizations can still be met by one single infant feeding scheme with the breastfed infant serving as the ideal example, or whether there should be different schemes for breastfed infants and for formula-fed infants (either fed follow-up formulas or YCF or CM) up to 24 months of age. The evaluated nutritional needs and recommendations are specific to protein, calcium, and iron due to their major impact on infants’ present and future health status.

## 3. Materials and Methods

In line with WHO recommendations [30], CF is supposed to start not before six months of age. Our nutritional analysis took into account only protein, iron, and calcium intake, not because all other nutrients are unimportant, but because these are the nutrients most frequently studied when assessing the nutritional status of children from 6 to 24 months of age in consideration of their impact on future health.

For checking the extent to which the nutritional needs of the schemes proposed are met, the European Food Safety Authority (EFSA) recommendations [31] were used because they are the most recent ones and those with the broadest list of references to official documents issued by other national and international agencies/organizations. The reference weight of infants was derived from WHO growth standards [32].

The weight calculation, reported in the tables, is based on the average of the 50th percentile weight at the youngest age and that at the oldest age in each of the two groups (6–12 months and 12–24 months). The resulting average weight was used to calculate the protein intake per kg of body weight. For the first two years of life, this simplified approach did not affect the results, as the authors also compared the intakes at both the lowest and highest weights and no significant difference was found in terms of results obtained (data not shown).

HM composition was taken from Picciano et al. [3]. Formula composition was calculated as the average composition of follow-up and YCF formulas available on the Italian market, respectively. Formulas with added biscuits or other foods/ingredients outside formula basic composition were not included. CM composition was taken from CREA (Italian Council for Agricultural Research and Analysis of Agricultural Economics) food composition tables [5]. Values of the average daily milk intake at different ages were taken from Dewey’s observational data [33]. The portion size of solid foods used for menus’ calculations come from portion sizes suggested in the recently published document on CF issued by the Italian Society of Preventive and Social Paediatrics (SIPPS) [34].

Daily menus were created with the same type and amount of solid food, leaving the amount of HM and formula unchanged in the first year of life. Unmodified CM was added to HM between 12 and 24 months of age. In the first age group, calcium- and iron-fortified cereals were used whereas, in the second year of life, common retail foods were used since fortified foods are less widely used. The vitamin D intake was not calculated in the two groups because 90% of the vitamin D present in the body comes from its production in the skin thanks to the sun’s rays and only 10% comes from food. So, for this irrelevant quantity the tables and the text have not been burdened.

All calculations made for the menus submitted are presented as Appendix A.

## 4. Results

Table 2, Table 3, Table 4, Table 5, Table 6 and Table 7 give the results based on the different intake values across the different menus.

Table 2 shows that the PRI for proteins is exceeded by both breastfed and formula-fed infants, although in formula-fed ones the excess is bigger, if the same solid food intake scheme is used (Δ + 0.2 g/kg/day at 6 months and +0.36 at 12 months with HM; +0.64 at 6 months and +0.8 at 12 months with a formula).

Table 3 and Table 6 show that the PRI for iron is almost met at the two considered ages only when a formula is used (Δ–1.2 and 0.8 mg/day at 6–8 and 18 months of age, respectively), but not if the infant is fed HM (Δ–7.6 and −5.4 mg/day at 6–8 and 18 months of age, respectively) or CM (Δ–5.2 mg/day at 18 months of age).

Table 4 and Table 7 show that calcium intakes, calculated in terms of Adequate Intake (AI), are met in the first age group in both HM and formula-based feeding, but in breastfed infants only thanks to the addition of calcium-fortified cereals, providing some extra 60 mg of calcium per day. It must be stressed that in formula-fed infants the use of calcium-fortified cereals is unnecessary since the AI is already met thanks to the formula (Δ +319 mg/day with fortified cereals, yet +259 with no surplus from cereals). As for the second year of life, calcium intake is never adequate in breastfed infants (Δ–309.5 mg/day), whereas formula-fed infants are close to the AI (Δ–22 mg/day), and in CM fed infants the AI is by far exceeded(Δ +159 mg/day). No correction was made, though, for the lower bioavailability of Ca from CM.

## 5. Discussion

### 5.1. Complementary Feeding between 6 and 12 Months of Age

#### 5.1.1. Proteins

Using a single solid food intake scheme, regardless of the type of milk feeding, protein intake always exceeds recommended levels. With breastfeeding, protein intake is closer to PRI values, but it increases considerably with formula feeding and even more with CM. Energy intake from protein is generally recommended not to fall below 6% and not to exceed 14% of the total daily calorie intake and safer if it stays between 8 and 12% [36]. With both HM and formula feeding, protein energy intake remains below 14% of the total calorie intake in the 6–8-month age group, reaching 6.8% and 9.2%, respectively. Additionally, in the 18-month age group with breastfeeding protein-derived energy intake is well below 14%, reaching only 9%, while with formula feeding it reaches about 12%, and almost 15% with CM. However, another more accurate way to assess intake adequacy, which is independent of energy intake and more personalised, consists of considering protein intake per kg and checking how much it departs from the PRI, which is 1.4 g/kg/day at 6–8 months and 1.03 g/kg/day at 18 months. Protein intake per kg is almost one and a half times the PRI in breastfed infants (1.5 g/kg/day) but increases and almost doubles to 1.9 g/kg/day in the case of formula-fed infants in the 6- to 8-month age group.

At 18 months, protein intake should be 1.03 g/kg/day as PRI/kg. In a HM based feeding, protein intake is 1.7 g/kg/day, i.e., one and a half times the recommended level; in the case of formula feeding, it is twice as high, i.e., 2.1 g/kg/day and, finally, when CM is used, it reaches three times the recommended level, i.e., 3 g/kg/day.

According to EFSA [35], it is not possible to establish a maximum daily intake level for protein and, for adults, an intake of twice the PRI is still considered safe. However, in the first two years of life, an excess of protein intake appears to be a risk factor for the development of obesity later in life [35].

Of all proteins, CM proteins have been shown to promote higher growth rates [37].

Thus, the use of a single CF scheme, based on breastfed infants, shows adequate protein intakes for breastfed infants and an excess for formula-fed ones, and even more so when CM is used, thus putting these infants at risk of developing obesity later in their life, a risk linked to both the amount and quality of their protein intake.

Consequently, when a single complementary feeding scheme is used for both breastfed and formula-fed infants, it becomes immediately evident that, while the protein intake of breastfed infants can be considered adequate, it turns out to be too high for infants fed follow-up formulas. Since follow-up formula-fed infants already have higher protein intakes than breastfed infants, it is not advisable to add protein-rich foods such as meat or fish or cheese to CF from the start (at 6 months). Quite the contrary, protein-rich foods are recommended from the start of CF for breastfed babies. When deciding between meat and fish, the latter should certainly be preferred for its higher AGE content and lower protein load.

If soy-based formulas are used, the situation does not change significantly since, due to the different amino acid composition, the protein content is even slightly higher than the CM-based formulas, while the iron content does not change significantly.

#### 5.1.2. Iron

The second half of the first year of life is the period with the greatest need for iron, both because of the increase in blood volume and because iron is needed for neuro-cognitive development.

The amount of iron absorption is determined mainly by the body’s iron reserves and the bioavailability of the iron taken in. The lower the iron reserve, the higher, within certain limits, the percentage of iron absorbed. Beyond the well-established bioavailability of haem iron (15–25%) [38] and non-haem iron (4–7%) [39], in the CF period it is also important to know the amount and bioavailability of the iron from iron-fortified cereals (3%) [40] and especially from human milk (34%) [41] and formulas (20%) [42]. With this information it is possible to check the adequacy of the total intake without invasive procedures and to detect possible risk factors.

As iron absorption increases when vitamin C intake is added to foods and is inhibited by the presence of Ca salts, tea, proteins, phytates, and Mn, attention should be paid, for example, to the Mn content of ready-to-eat cereals, which often contain Mn levels between 1 and 4 mg/100 g [43].

Meeting iron intake needs in this period of life is not easy and, to increase iron intake, despite the (yet poor) bioavailability of Fe with some specific iron-supplemented infant foods other than formulas, the use of the latter can still be helpful. It is important not to expose the infant to the risk of an inadequate iron intake and, therefore, of developing Iron Deficiency (ID) and Iron Deficiency Anaemia (IDA), as iron therapy can only remedy haematological abnormalities, but not brain damage, which is irreversible [44,45,46].

Table 3 indicates that from 6 months onwards a breastfed baby (despite the high bioavailability of iron in HM) will also need to be given Fe-rich foods to help with its absorption or to receive Fe supplements. However, any increase in the portion of meat to increase Fe intake will not be helpful, as even tripling the amount of meat may not achieve the PRI value for Fe, while increasing protein intake excessively.

#### 5.1.3. Calcium

According to EFSA, Ca AI is 280 mg [35]. BM has a lower Ca content (23 mg%) than follow-up formulas (70 mg%), but its 50% bioavailability is much higher than that of follow-up formulas where it reaches only 30–35%.

For adequate intakes to be achieved, iron-fortified cereals are not critically necessary for formula-fed infants, whereas they may be useful in the case of breastfed infants, even though HM calcium has a higher bioavailability than other foods.

### 5.2. Complementary Feeding between 12 and 24 Months of Age

In the second year of life, the number of questions to the paediatrician to receive information, clarifications, and advice on a child’s diet reduces considerably, while the risk of an incorrect (by over- or under-coverage) coverage of nutritional needs of the baby is still high, in a period of life that is still particularly sensitive in terms of long-term outcomes. Foods rich in added sugar and salt are easily introduced instead of healthier foods such as fruit and vegetables. The reasons for such behaviour may be found both in the widespread advertising of infant food, as well as in the widespread belief/desire of mothers (and grandmothers) that their child is “all grown up” and can and should eat much more foods suitable for older children.

The World Health Organisation (WHO), in its paper on CF of the breast-fed child [30], states that breastfeeding can continue into the second year of life but gives no indication about what to do should HM no longer be available. Again the WHO, in its now quite outdated paper on the feeding of non-breastfed babies [47], states that “Acceptable milk sources include full-cream animal milk (cow, goat, buffalo, sheep, camel), Ultra High Temperature (UHT) milk, reconstituted evaporated (but not condensed) milk, fermented milk or yogurt, and expressed breast milk … Commercial infant formula is an option when it is available, affordable, can be safely used, and provides a nutritional or other advantage over animal milk … Semi-skimmed milk may be acceptable after 12 months of age”. While this paper is primarily addressed to developing countries, it cannot be overlooked that different animals produce significantly diverse types of milk, and more importantly, they produce types of milk that differ from HM. Nor does this document help in the choice, after 12 months of age, of a milk source for non-breastfed children living in industrialised countries. Furthermore, the inclusion of even semi-skimmed CM at this age may expose infants, especially those from families with disadvantaged educational and socio-economic backgrounds, to a significant reduction in terms of total daily energy intake.

In a paper published in 2013, EFSA [48] states that “No unique role of young-child formulae with respect to the provision of critical nutrients in the diet of infants and young children living in Europe can be identified, so that they cannot be considered as a necessity to satisfy the nutritional requirements of young children when compared with other foods that may be included in the normal diet of young children (such as breast milk, infant formulae, follow-on formulae and cow‘s milk)”. Furthermore, in the same paper, EFSA recognises that “However, at this age (after the first year of) cow‘s milk consumption is no longer discouraged and no recommendations for replacement of this food category by other alternatives exist from medical societies at European level”.

Therefore, EFSA also lumps together HM, starting and follow-up formulas and even CM as alternatives to each other, without considering their significantly different nutritional properties.

Finally, ESPGHAN [49] states that “based on available evidence there is no necessity for the routine use of YCF in children from 1-3 years of life, but they can be used as part of a strategy to increase the intake of iron, vitamin D and n-3 PUFA and decrease the intake of protein compared to unfortified cow’s milk. Follow-on formulae can be used for the same purpose”.

In conclusion, the leading scientific societies and international institutions (WHO, EFSA) give freedom when it comes to the consumption of whatever infant formula or milk may be available, but no data are available to confirm the nutritional adequacy of the different types of milk used to supplement solid food consumption between one and two years of age. However, it should be emphasised that beyond one year of age, milk (of whatever kind) is no longer the main food to be supplemented, but rather milk could be said to be the food that complements solid food intake. Therefore, the portions of the latter should be adjusted to the composition of the different types of milk. Even so, given that during this period of life milk shares energy and nutrient intakes with many other foods, the total nutrient intake may vary greatly, depending on whether one chooses HM, CM, or YCFs.

One issue arises from the fact that YCFs are not regulated (yet) on a national or European basis in their nutrient and energy composition, thus reflecting a heterogeneous group of formulations, some of which are more targeted to the specific nutritional needs of children of this age, while others appear to be just a marketing gimmick [50].

In the 2018 ESPGHAN paper [49], it is further stated that there are no obstacles to the use of follow-up formulas also in feeding children beyond one year of age and that, therefore, it may not be necessary to define special regulations and values for YCF.

Finally, any regulation of YCF composition should take into account that the regular intake of food for the whole family may differ across European countries and families and the reduction in milk intake depends very much on the child’s diet preferences and the family’s eating habits. Therefore, when choosing to use one of these formulas, paediatricians should be able to assess its nutritional adequacy and usefulness within the context of the whole child’s diet.

The most common nutritional problems in the diets of children between 12 and 24 months of age are likely to be related to an insufficient intake of Fe and an excessive intake of protein and Na. The problem of excessive protein intake is the most studied, and there is some limited evidence [51] that excessive protein intake in the first two years of life promotes the development of obesity later in life.

Of all proteins, those from milk and dairy products appear to play the most important role, not only because of their specific quality, but also because, although meat, fish, and eggs contain higher protein percentages, the overall amounts of milk consumed at this age are still likely to make milk the most important source of the total protein intake.

While breastfeeding has been shown to be a protective factor for the development of obesity, the intake of unmodified CM, even at this age, is definitely a risk factor not only for obesity but also for iron deficiency on the grounds that Fe is almost totally absent in CM. YCFs are at lower risk than CM for both diseases, given their lower protein content and higher Fe content.

However, given the great variability in the eating habits of families, examples of how to meet the recommendations for protein, Fe, and Ca are given above (Table 5, Table 6 and Table 7) for an 18-month-old child fed HM, fed YCF, or fed unmodified CM, but with the same portions of both milk and solid foods and with the same energy intake. Protein intake is the lowest with HM (1.7 g/kg), it increases to 2.1 g/kg with YCF, but reaches 2.9 g/kg with CM, almost tripling the PRI for protein. Fe intake is inadequate with both HM and CM, whereas it is met with YCF. Finally, Ca intake is met with both CM and YCF, whereas with breastfed infants it may be useful to use Ca-fortified cereals.

These schemes are based on observed intakes of healthy infants and also correspond, as far as CM is concerned, to the proposal of several international societies/organizations which, in view of the possible risks linked to excessive consumption of CM even in the second year of life, recommend that the daily intake of CM should not exceed 500 mL/day. As can be inferred from the tables presented here, this limit does not remedy excessive protein intake.

Strengths and limitations of the study. Limitations: The menus used for calculations may not represent the variety of menus offered to children in a certain community, and they may not respect the use of local foods. Our aim, however, is to show that the use of one single model of introducing complementary foods is not adequate for both breastfed and formula-fed infants, irrespective of the menus used. Since we want to stress the principle that CF should be differentiated in breastfed vs. formula-fed infants, we kept the menus as simple as possible; for this reason, we avoided considering specific eating habits. In any case, indeed, the use of local foods would be similar in both groups of children and will result in the same nutritional problems. Another limitation of this study is that we limited the nutritional analysis to specific types of milk, i.e., we did not consider a diet with simultaneous intake of different types of milk (HM, formula, or CM). The combinations of mixed feeding are innumerable and illustrating all the different combinations would have made this paper excessively long. Hence, it is of paramount importance for the paediatrician to evaluate thoroughly what kind(s) of milk the single infant takes and in which quantity. This way the health professional will be able to tailor the infant’s diet to his/her real needs without exceeding them or, on the contrary, staying below them. Strengths: This is the first study, to our knowledge, that analyses the intakes of certain nutrients in breastfed vs. formula-fed infants using a single solid food intake pattern. This article can be helpful in providing better advice on offering solid foods for both breastfed and formula-fed infants to allow them the best possible growth.

## 6. Conclusions

The type of milk (or formula) the infant is fed from six to twelve months of age should determine the quality of the solid foods the infant in question is offered.

Exclusively breastfed infants should, therefore, be offered protein-rich foods such as meat, fish, pulses, cheese, and eggs from the beginning of the CF period. However, as these foods, in adequate quantities, do not meet iron and calcium requirements, the use of fortified cereals may be helpful.

Exclusively formula-fed infants, on the other hand, should not be offered such naturally rich or enriched foods from the beginning, since formula milk contains more than enough protein, iron, and calcium. On the contrary, formula-fed infants should be offered a greater variety of fruit and vegetables from the beginning to promote the development of their taste for different flavours, given that the formula flavour experience is more monotone.

In the second year of life (12 to 24 months), if HM is not available, a YCF can more easily meet the age-related nutrient intake recommendations than unmodified CM. Additionally, in this age group, it is important to avoid excessive protein intake and insufficient iron and calcium intake.

In conclusion, the statements of ESPGHAN [49] on the futility of proposing complementary feeding schemes that vary according to the type of milk feeding (because of fear of possible confusion on the part of adults) do not allow to fully meet the needs and nutritional recommendations issued by the main Agencies/Organizations/Societies for all children within this age group.

Furthermore, the proposals of WHO [47] and EFSA [48] on the possibility of using any type of milk from mammals available in the different regions of the world for infants in the 12- to 24-month age group, while understandable from the point of view of respecting local culture and local supply, should be accompanied by a few simple dietary recommendations to help meet the nutritional requirements of infants as best as possible, given the different compositions of milk from different mammals.

Since nutrition in this period of life is of crucial importance for the development of adult life and especially for a sound psycho-neuro-motor development of an individual, all existing scientific knowledge must be used at best to allow each child to reach his/her full genetic potential.

## Figures and Tables

**Table 1 nutrients-13-03756-t001:** Composition of human breast milk, follow-up formulas, young child formulas (YCFs), and cow milk (CM).

Food	mL	kcal	Proteins g	Total Fats g	Saturated Fats g	Carbohydrates g	Simple Sugars g	Iron mg	Calcium mg
Human milk (HM) *	100	68.0	0.90	3.50	1.57	8.00	8.00	0.06	23
Follow-up formula ^	100	67.5	1.41	3.21	1.26	8.15	6.06	0.99	70
Young child formula (YCF) ^	100	60.6	1.66	2.60	0.85	6.10	§	1.00	82
Cow milk (CM) °	100	64.0	3.30	3.60	2.10	4.90	4.90	0.10	119

* Values from [3]. ^ Edited from [4], for formulas available in Italy. ° From [5]. § Beside lactose, YCFs may contain, in varying and not always declared proportions, complex carbohydrates (maltodextrins, starch, cereal flour) as well as simple carbohydrates (sucrose, dextrose, glucose, fructose), thus it is not possible to calculate any average content of simple sugars.

**Table 2 nutrients-13-03756-t002:** Different protein intake for a 6–8-month-old infant when breastfed or formula-fed, with a single solid food intake scheme, assuming a body weight between 7.3 and 8.6 kg [32] and a PRI for protein of 1.3 g/kg/day [35]. The amount of milk * is defined according to Dewey’s observational data [7].

	Proteins in g	
Food in Portions/Day	HM	Follow-Up Formula
Milk 688 * mL	6.2	10.1 (average content)
Cereals 25 g (average content)	2.5	2.5
Extravergin olive oil 10 g		
Vegetables 20 g (average content)	0.5	0.5
Veal 10 g	2.1	2.1
Fruits 40 g (average content)	0.3	0.3
Total g	11.8	15.5
g/kg/day	1.5	1.94
PRI	1.3 g/kg/day at 6 months 1.14 g/kg/day at 12 months	

**Table 3 nutrients-13-03756-t003:** Different iron intake for a 6–8-month-old infant weighing between 7.3 and 8.6 kg [32] when breastfed or formula-fed, considering only one single complementary feeding scheme and an iron PRI of 11 mg/day [35]. The amount of milk * is defined according to Dewey’s observational data [7].

	Iron in mg	
Food in Portions/Day	HM	Follow-Up Formula
Milk 688 * mL/day	0.4	6.8 (average content)
Iron-fortified cereals 25 g (average content)	2.4	2.4
Extravergin olive oil 10 g	0.0	0.0
Vegetables 20 g (average content)	0.2	0.2
Veal 10 g (average content)	0.2	0.2
Fruit 40 g (average content)	0.2	0.2
Total mg/day	3.4	9.8
PRI for Fe	11 mg/day	

**Table 4 nutrients-13-03756-t004:** Different Ca intake for a 6–8-month-old infant weighing between 7.3 and 8.6 kg [32] when breastfed or formula-fed, considering only one single complementary feeding scheme and an AI for Ca of 280 mg/day [34]. Milk quantity * is defined according to Dewey’s observational data [7].

	Ca in mg	
Food in Portions/Day	HM	Follow-Up Formula
Milk 688 * mL	158	482 (average content)
Ca fortified cereals 25 g (average content)	60	60
Extravergin olive oil 10 g	0	0
Vegetables 20 g (average content)	39	39
Veal 10 g	0	0
Fruits 40 g (average content)	18	18
Total mg/day	275	599
Calcium Adequate Intake (AI)	280 mg/day	

**Table 5 nutrients-13-03756-t005:** Different protein intakes for an infant of around 18 months of age weighing approximately 10.2–10.9 kg [32] when breastfed, fed with YCF or CM, using a single scheme of intake of solid foods commonly used by the family, and the same portion of milk, and considering a PRI for protein of 10.1–13.7 g/day (1.0–1.3 g/kg) [34]. The amount of milk * is defined according to Dewey’s observational data [7].

Food	Proteins in g per Portion		
Portions/Day	HM	Young Child Formula (YCF)	Cow Milk (CM)
Milk 488 * mL	4.4	8.1	17.1 (average content)
Pasta 30 g	3.3	3.3	3.3
Chicken breast 20 g	4.7	4.7	4.7
Extravergin olive oil 20 g	0.0	0.0	0.0
Vegetables 60 g (average content)	0.8	0.8	0.8
Rice 30 g	2	2	2
Peas 30 g	1.6	1.6	1.6
Fruit 150 g (average content)	0.6	0.6	0.6
Total in g	17.4	21.1	30.1
g/kg/day	1.7	2.1	3
PRI	1.03 g/kg/day		

**Table 6 nutrients-13-03756-t006:** Different iron intakes for an infant of around 18 months of age weighing approximately 10.2–10.9 kg [32] when breastfed, fed with YCF or CM, using a single scheme of intake of solid foods commonly used by the family, and the same portion of milk, and considering a PRI for iron of 8 mg/day [3,4,5,7,35]. The amount of milk * is defined according to Dewey’s observational data [7].

Food	Iron in mg		
Portions/Day	HM	YCF	CM
Milk 488 * mL	0.3	4.9 (average content)	0.5 (average content)
Pasta 30 g	0.4	0.4	0.4
Chicken breast 20 g	0.1	0.1	0.1
Extravergin olive oil 20 g	0.0	0.0	0.0
Vegetables 60 g (average content)	0.4	0.4	0.4
Rice 30 g	0.2	0.2	0.2
Peas 30 g	0.6	0.6	0.6
Fruit 150 g (average content)	0.6	0.6	0.6
Total in mg/day	2.6	7.2	2.8
PRI for Fe	8.0 mg/die		

**Table 7 nutrients-13-03756-t007:** Different Ca intakes for an infant of around 18 months of age weighing approximately 10.2–10.9 kg [32] when breastfed, fed with YCF or CM, using a single scheme of intake of solid foods commonly used by the family, and the same portion of milk, and considering a Ca AI of 450 mg/day [3,4,5,7,35]. The amount of milk * is defined according to Dewey’s observational data [7].

Food	Ca mg		
Portions/Day	HM	YCF	CM
Milk 488 * mL	112	400 (average content)	581 (average content)
Pasta 30 g	6.6	6.6	6.6
Chicken breast 20 g	0.8	0.8	0.8
Extravergin olive oil 20 g	0.0	0.0	0.0
Vegetables 60 g (average content)	0.4	0.4	0.4
Rice 30 g	7.2	7.2	7.2
Peas 30 g	6	6	6
Fruit 150 g (average content)	7.5	7.5	7.5
Total mg/day	140.5	428	609
Calcium AI	450 mg/day		

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
