# Peer review of "Breastfed and Formula-Fed Infants: Need of a Different Complementary Feeding Model?"

_nutrients, 2021, doi:10.3390/nu13113756_

Round 1
Reviewer 1 Report
This interesting study assessed if recommendations for complementary feeding from scientific bodies and societies, specifically concerning protein, calcium and iron, should be different in infants either breastfed or formula fed up to 24 months of age.
Some issues need to be addressed:
- When stating the Objectives, the authors should be specified that the evaluated nutritional needs and recommendations are restricted to protein, calcium and iron.
- Strengths and limitations of the study should be acknowledged in Discussion section. For instance, daily assumed menus that were used for calculations may not represent the variety of menus offered to children in a certain community and even less representative in different communities with expected different habits, and this should be acknowledged as a limitation. In other words, fixed menus do not consider local culture and food supply, as the authors emphasize (lines 405-410).
- Abbreviations should be explained in extent when firs used in the text: PRI (line32), CM (line 49), inadequately explained in extent afterwards (line 53), ID (276), IDA (line 277)
Author Response
First of all we want to thank the 3 reviewers for their very nice words regarding our paper: we really appreciate it. Furthermore, all the comments made us understand that they are truly experts in the field of paediatric nutrition and receiving positive comments from these experts honours us.
Reviewer n 1
This interesting study assessed if recommendations for complementary feeding from scientific bodies and societies, specifically concerning protein, calcium and iron, should be different in infants either breastfed or formula fed up to 24 months of age.
Some issues need to be addressed:
- “When stating the Objectives, the authors should be specified that the evaluated nutritional needs and recommendations are restricted to protein, calcium and iron.”
We thank the reviewer for the wise remark. It was actually omitted by us, so we added what was suggested using the reviewer's sentence (if she/he allows us). The section now ends with the statement “The evaluated nutritional needs and recommendations are restricted to protein, calcium and iron for their major impact on infants’ present and future health status.”
- Strengths and limitations of the study should be acknowledged in Discussion section. For instance, daily assumed menus that were used for calculations may not represent the variety of menus offered to children in a certain community and even less representative in different communities with expected different habits, and this should be acknowledged as a limitation. In other words, fixed menus do not consider local culture and food supply, as the authors emphasize (lines 405-410).
Thanks a lot for this comment. We totally agree with the reviewer that the menus used for calculations may not represent the variety of menus offered to children in a certain community, and that they may not respect the use of local foods. Our aim, however, was slightly different: we wanted to show that the use of one single model of introducing complementary foods is not adequate for both breast-fed and formula-fed infants, irrespective of the menus. Since we want to stress the principle that complementary feeding must be differentiated in breastfed vs. formula fed infants, we kept the menus as simple as possible; for this reason we avoided considering specific eating habits. In any case, the use of local foods would be similar in both groups of children and will result in the same nutritional problems. Taking into account the reviewer’s important comment, however, we added a phrase in the discussion section acknowledging this limitation and our explanation for it.
“Strengths and limitations of the study. Limitation. The menus used for calculations may not represent the variety of menus offered to children in a certain community, and they may not respect the use of local foods. Our aim, however, is to show that the use of one single model of introducing complementary foods is not adequate for both breastfed and formula-fed infants, irrespective of the menus used. Since we want to stress the principle that complementary feeding must be differentiated in breastfed vs. formula fed infants, we kept the menus as simple as possible; for this reason, we avoided considering specific eating habits. In any case, indeed, the use of local foods would be similar in both groups of infants and will result in the same nutritional problems. Strengths. This is the first study, to our knowledge, that analyses the intakes of certain nutrients in breast-fed vs. formula-fed infants using a single solid food intake pattern. This article can be helpful in providing better advice on offering solid foods for both breastfed and formula fed infants to allow them the best possible growth.
Abbreviations should be explained in extent when first used in the text: PRI (line32), CM (line 49), inadequately explained in extent afterwards (line 53), ID (276), IDA (line 277)
Many thanks for the remark. We apologize for these inaccuracies, which have now been corrected. The abbreviation PRI at line 32 has been explained: Population Recommended Intake (PRI), as well as for iron deficiency (ID, line 276) and iron deficient anaemia (IDA, line 277). “Cow milk” written in extent in line 49 has been kept as the first appearance in the text and 53 has been eliminated.
Reviewer 2 Report
Dear Authors
Thank you for this much needed paper. I have a few comments..
1) Follow-on formula is not relevant in the weaning period, I would remove that and replace that with a US infant formula as they contain even more iron the EU formulas.
2) as Portion sizes for infants have not been published and only suggested in the draft US dietary guidelines, where did you get your portion sized from?
3) can you use the portion sizes for EU infants and EU formulas and then US portion sized and US formulas in the revised version?
Author Response
First of all we want to thank the 3 reviewers for their very nice words regarding our paper: we really appreciate it. Furthermore, all the comments made us understand that they are truly experts in the field of paediatric nutrition and receiving positive comments from these experts honours us.
Reviewer n2
Dear Authors
Thank you for this much needed paper. I have a few comments..
1) Follow-on formula is not relevant in the weaning period, I would remove that and replace that with a US infant formula as they contain even more iron the EU formulas.
Thanks for the remark. We apologise but we cannot respond positively to the reviewer’s suggestion, since US formulas cannot be sold in Europe (where starting and follow-on formulas are ruled in strict details by UE regulations). As a consequence, not only we don’t have experience of their use, but also, by using US standards, we would have failed in our purpose, which is to stress the inappropriateness of the statement issued by ESPGHAN (the European branch of the WSPGHAN) about the uselessness of having different schemes for the two kinds of infants, breastfed vs. formula-fed.
2) as Portion sizes for infants have not been published and only suggested in the draft US dietary guidelines, where did you get your portion sized from?
Many thanks for the comment. The portions of solid foods used here derive from the portions suggested by the document on complementary feeding of the Italian Society of Preventive and Social Paediatrics, very recently published. Following the correct question of the reviewer we have added the bibliographic reference. We want to stress, however, that portion size standards used, if different, may impact on calculations, but not at all on the existence of a profound difference between breastfed and formula-fed infants.
3) can you use the portion sizes for EU infants and EU formulas and then US portion sized and US formulas in the revised version?
We thank the reviewer for the suggestion. We apologise, once again, but we cannot respond positively. In our research we used the average composition of both follow-on formulas and YCFs sold in Italy, which – as said before – follow the very strict European legislation. Also for baby food we used the average nutritional composition of the products sold in Italy, but which are, once again, subjected to strict both national and European regulations. We have no experience with the products sold in US and we would be very happy if this reviewer's excellent suggestion was grasped by paediatric nutrition researchers living in the US, and adapted to their different standards, as they could do so much more competently than we, unfamiliar with these products, could.
4) Moderate English changes required
We apologize for not to having been able to write the text of our article in perfect English. We have entrusted the translation to Dr Lucia Sollecito, translator and simultaneous interpreter in the medical field. It is possible that the translator used a UK English style, slightly different from the English used in the US. However we have changed several phrases in the text.
Reviewer 3 Report
This is an interesting and relevant paper in which the authors seek to show that recommendations for complementary foods from 6 to 24 months of life need to take into account the type of milk feeding to achieve appropriate protein, calcium , and iron intake and that failure to do so may contribute to negative health outcomes, with obesity and effects of iron deficiency highlighted. They challenge the view of ESPGHAN which, although recognizing the differences in "composition and health effects" between breastmilk and formulas, felt that separate recommendations for complementary foods were not practical and could confuse parents. This paper claims that this conclusion is not evidence-based and their study provides a rational to counsel parents in appropriate complementary food choices beyond 6 months of age.
Overall, the paper is well written, presented, and discussed. The introduction is quite comprehensive and well referenced. The age span is divided sensibly into two periods, 6 to 12 months and 12 to 24 months taking into account the WHO recommendations for infant feeding. The data focuses on the midpoint of each age period in presenting nutrient intake using a preset daily menu of typical complementary foods and type and volume of milk (breastmilk, cows milk, and formulas). The tables are informative, and could provide further insight if they included the bioavailability of calcium and iron in the different schemes as opposed to only in the text. I was curious as to why vitamin D was not included in the nutrients studied. It is certainly a nutrient of concern and is measured. This should be explained. Also fat composition was omitted which may not be measured frequently in infants but still has implications for health outcomes
I understand the reason for using preset menus in the methodology but in the discussion there was no mention that different milk feeding modes may also influence the type and quality of complementary foods chosen by parents, such as home prepared foods, and this may affect nutrient intake. There is good discussion about cultural factors and milk/ formulas available as well as concerns about the lack of information on nutrients in other mammalian milk . The conclusion provides summary and practical recommendations.
Comments/Suggestions:
The authors should take care to use the terms "complementary" feeding and "supplementing" correctly throughout. e.g line 23
In 2021 there is a call to use the term "human milk" (HM) instead of breastmilk - a possible consideration.
Use of abbreviations - There were quite a few abbreviations that were not written in full the first time they appear. This was frustrating to a reader not familiar with the abbreviations used. Examples: PRI (line32), ESPGHAN (Iine 34) - both in the abstract which might be okay per the editors but then they are not spelled out the first time in the text, WHO (line 142),EFSA (line 148), YCF (line 160), AGE (line255), ID (line 276) and IDA (line 277)
Lines 32-33. This sentence needs revision . "Iron PRI and calcium. Adequate intakes ....." Suggest changing to "Adequate intake of iron PRI and calcium are met ......" Also this is the first time PRI is used . Please write it out the first time.
Line 94 -Change aminoacidic to aminoacid
Lines 98-99: Suggestion to revise this sentence to .... different effects on body composition from the very first months of life into adulthood and may influence long term health outcomes
In lines 104-107, there is a real need to revise or clarify this paragraph as it can be interpreted that a greater amount of visceral fat" is a protective factor in the development of metabolic factors later in life rather than a risk factor.
Lines 113-116: The higher fat composition in hind-milk versus fore-milk in breastfed infants may also contribute to appetite regulation in addition to hormones.
Lines 121: ..."the two different lactation patterns".... I think this is an error and is not two different breastfeeding pattens but rather refers to the two feeding patterns (breastmilk and formulas).
Lines 152-157: This paragraph is confusing. ....lowest average weight at the youngest age ......etc doesn't make sense to this reader. Is the weight calculation based on the average of the 50th percentile weight at the youngest age and that at the oldest age in each of the 2 groups (6-12 months and 12-24 months). This is what I concluded. Please make this clearer.
Line 181: ."...the use of calcium fortified cereals is useless" Suggest replacing "useless" with "unnecessary"
In the conclusions, consider including iron supplementation in the options for meeting iron intake requirements in breastfed infants as well as iron rich foods and sources of Vitamin C to increase absorption. Also Iron supplementation in CM-fed older infants and Vit C sources may also help to prevent iron deficiency without increasing protein intake.
Young Child Formula is suggested to meet adequate and appropriate intake of iron and protein in the older non-breastmilk fed group. Cost is not mentioned although availability is discussed in general. It would also be more comprehensive in the discussion to include plant-based milk such as soy milk and possible concerns.
One difficulty in recommendations is that many children in the 6 to 24 month age group are not fed from a single milk source - such as mixed human milk and formula or human milk and CM which complicates clear messaging about complementary foods. This is not addressed specifically but further highlights the need to look at the milk source and also understand the implications for achieving appropriate nutrient intake when advising about complementary foods.
I enjoyed reading and reviewing this paper which is thought provoking and helps to fill the gap in evidence-based knowledge to guide pediatricians and others involved in the counseling of parents and also those who advocate or set guidelines for adequate nutrition beyond 6 months in both breastmilk-fed and non-breastmilk-fed infants.
Author Response
First of all we want to thank the 3 reviewers for their very nice words regarding our paper: we really appreciate it. Furthermore, all the comments made us understand that they are truly experts in the field of paediatric nutrition and receiving positive comments from these experts honours us.
Reviewer n3
This is an interesting and relevant paper in which the authors seek to show that recommendations for complementary foods from 6 to 24 months of life need to take into account the type of milk feeding to achieve appropriate protein, calcium , and iron intake and that failure to do so may contribute to negative health outcomes, with obesity and effects of iron deficiency highlighted. They challenge the view of ESPGHAN which, although recognizing the differences in "composition and health effects" between breastmilk and formulas, felt that separate recommendations for complementary foods were not practical and could confuse parents. This paper claims that this conclusion is not evidence-based and their study provides a rational to counsel parents in appropriate complementary food choices beyond 6 months of age.
Overall, the paper is well written, presented, and discussed. The introduction is quite comprehensive and well referenced. The age span is divided sensibly into two periods, 6 to 12 months and 12 to 24 months taking into account the WHO recommendations for infant feeding. The data focuses on the midpoint of each age period in presenting nutrient intake using a pre-set daily menu of typical complementary foods and type and volume of milk (breast-milk, cow milk, and formulas). The tables are informative, and could provide further insight if they included the bioavailability of calcium and iron in the different schemes as opposed to only in the text.
Thanks for the remark. We are happy that the reviewer has considered the issue of the bioavailability of iron and calcium in the different menus because it is an aspect that we have long considered whether to include or not. Our final decision was then not to include the percentage of bioavailability of iron and calcium because bioavailability depends on many factors such as the amount of protein, the chemical form of the mineral, the type of fibre, the presence of saturated FFA, and so on. Moreover, international institutions such as ESPGHAN, EFSA and WHO limit themselves to recommending a total intake of these minerals, considering the fact that in some foods the bioavailability is higher and in other foods is lower, but without going into details, possibly – we infer – because the variety of the whole daily diet averages it.
I was curious as to why vitamin D was not included in the nutrients studied. It is certainly a nutrient of concern and is measured. This should be explained.
It is a wise observation, but, after discussing the topic, we decided not to include vitamin D intake in the menu analysis because 90% of the vitamin D present in the body comes from its production in the skin thanks to the sun's rays and only 10% comes from food. It seemed irrelevant to report this data as well so as not to burden tables and text. However, we have included the explanation of this interesting comment in the text of the "materials and methods" section.
Also fat composition was omitted which may not be measured frequently in infants but still has implications for health outcomes.
Thanks for the comment. We agree that lipids’ supply is an essential aspect in nutrition from the very beginning of life. However, not all formulas (and especially the YCF) have the same composition (in absolute terms and percentages) and quality of lipids and above all, as for human milk, the lipid part is the one most subject to changes, due to the type of diet followed from the mother before and during breastfeeding. It seemed to us that we could arrive at non incontrovertible conclusions and give indications that are not supported by sufficient evidence.
I understand the reason for using pre-set menus in the methodology but in the discussion there was no mention that different milk feeding modes may also influence the type and quality of complementary foods chosen by parents, such as home prepared foods, and this may affect nutrient intake. There is good discussion about cultural factors and milk/ formulas available as well as concerns about the lack of information on nutrients in other mammalian milk . The conclusion provides summary and practical recommendations.
Since there is another comment on this topic below, we will respond to this point further in our reply.
Comments/Suggestions:
The authors should take care to use the terms "complementary" feeding and "supplementing" correctly throughout. e.g line 23
Thanks a lot for this lexical help. The word “supplement” has been changed in “complement”.
In 2021 there is a call to use the term "human milk" (HM) instead of breastmilk - a possible consideration.
We agree to use, wherever possible, the definition “human milk” instead of “breast-milk” in the text.
Use of abbreviations - There were quite a few abbreviations that were not written in full the first time they appear. This was frustrating to a reader not familiar with the abbreviations used. Examples: PRI (line32), ESPGHAN (Iine 34) - both in the abstract which might be okay per the editors but then they are not spelled out the first time in the text, WHO (line 142),EFSA (line 148), YCF (line 160), AGE (line255), ID (line 276) and IDA (line 277)
Thanks for the remark The lack of explanation of some acronyms was also pointed out by the first reviewer. According to your right suggestion we explained the acronyms not only in the abstract but also in text and tables when the acronym was first used.
Lines 32-33. This sentence needs revision . "Iron PRI and calcium. Adequate intakes ....." Suggest changing to "Adequate intake of iron PRI and calcium are met ......" Also this is the first time PRI is used . Please write it out the first time.
We have re-written the sentence considering, in addition to the lack of an explanation of the acronym, also the different nutritional meanings of PRI Population Recommended Intake and Adequate Intake.
Line 94 -Change aminoacidic to amino acid
We apologize for these minor inconveniences we have caused to all reviewers. We changed the word “aminoacidic” into “amino acid” wherever it appears in the text
Lines 98-99: Suggestion to revise this sentence to .... different effects on body composition from the very first months of life into adulthood and may influence long term health outcomes
Thanks for the remark and for the suggested phrase. The sentence was changed as kindly suggested by the reviewer
In lines 104-107, there is a real need to revise or clarify this paragraph as it can be interpreted that a greater amount of visceral fat" is a protective factor in the development of metabolic factors later in life rather than a risk factor.
Re-reading the paragraph, we agree that it is highly unclear, thanks for noticing it. The clause “which seems to be a protective factor in the development of metabolic changes at later ages” has now been changed into “with the first condition (more subcutaneous than visceral fat) seeming to be a protective factor in the development of metabolic changes at later ages”
Lines 113-116: The higher fat composition in hind-milk versus fore-milk in breastfed infants may also contribute to appetite regulation in addition to hormones.
Totally correct, thanks for adding this point. Confiding in the reviewer’s benevolence, we have added his/her exact sentence in the point indicated.
Lines 121: ..."the two different lactation patterns".... I think this is an error and is not two different breastfeeding pattens but rather refers to the two feeding patterns (breastmilk and formulas).
Thanks a lot, error emended.
Lines 152-157: This paragraph is confusing. ....lowest average weight at the youngest age ......etc doesn't make sense to this reader. Is the weight calculation based on the average of the 50th percentile weight at the youngest age and that at the oldest age in each of the 2 groups (6-12 months and 12-24 months). This is what I concluded. Please make this clearer.
The sentence the reviewer used to check his/her understanding of the text is exactly what we wanted to explain. Trusting in the reviewer’s permission, we transfer it entirely to the text.
Line 181: ."...the use of calcium fortified cereals is useless" Suggest replacing "useless" with "unnecessary"
The suggested change has been made, Thanks.
In the conclusions, consider including iron supplementation in the options for meeting iron intake requirements in breastfed infants as well as iron rich foods and sources of Vitamin C to increase absorption. Also Iron supplementation in CM-fed older infants and Vit C sources may also help to prevent iron deficiency without increasing protein intake.
Thank for this extremely important point. We fully agree that the use of foods rich in vitamin C could increase the bioavailability of iron and therefore its absorption, but this would happen in each group following a different menu and therefore the difference would still be maintained. Finally, we honestly do not believe that the addition of vitamin C can increase the bioavailability of the minimum quantities of iron present in the HM or CM as much as to satisfy the needs for daily iron in this precise period of life.
We certainly agree that iron pharmacological supplementation may help to reach iron nutritional needs, but, in truth, this can modify the intestinal microbiota more than can happen with a formula or a baby food supplemented with iron Simonyté Sjödin K, Domellöf M, Lagerqvist C, et al. Administration of ferrous sulfate drops has significant effects on the gut microbiota of iron-sufficient infants: a randomised controlled study. Gut. 2018; pii: gutjnl-2018-316988. Since it has been shown that the microbiota is extremely important in modelling the immune response and the state of health in general we did not want to risk a suggestion that could be not positive in many respects.
Young Child Formula is suggested to meet adequate and appropriate intake of iron and protein in the older non-breastmilk fed group. Cost is not mentioned although availability is discussed in general. It would also be more comprehensive in the discussion to include plant-based milk such as soy milk and possible concerns.
We have not included soy-based formulas because, in Europe, these should (at least theoretically) only be used in case of allergy to cow's milk proteins, while our article addresses the problem of CF in healthy children. Moreover, even soy-based formulas must comply with the European legislation in this regard and therefore there are only few differences between the two formulations (soy-based vs CM-based). However, we have included a sentence in the discussion underlining the fact that, in case of use of soy-based formulas, the situation does not change because, due to the different amino acid composition, the protein content is even slightly higher than the CM-based formulas, while the iron content does not change significantly.
One difficulty in recommendations is that many children in the 6 to 24 month age group are not fed from a single milk source - such as mixed human milk and formula or human milk and CM which complicates clear messaging about complementary foods. This is not addressed specifically but further highlights the need to look at the milk source and also understand the implications for achieving appropriate nutrient intake when advising about complementary foods.
Thank you very much for this particularly important comment. In fact we considered the problem of mixed milk feeding but for the sake of schematization and simplicity we finally preferred to take into account only the extreme situations (breast-feeding vs. formula feeding). However, we have now added to the discussion, and in particular in the limitations’ section, a sentence about the possibility of having a mixed diet in various ways and on how to deal with the problem. We report here the changes made: “Another limitation of this study is that we limited the nutritional analysis to "extreme" situations, i.e., we did not consider a diet with simultaneous intake of different types of milk (maternal, formula, or CM). The combinations of mixed feeding are innumerable and illustrating all the different combinations would have unnecessarily lengthened the article. For a correct nutritional approach it is necessary to know the percentages of the various types of milk taken in order to have the best approach in choosing complementary foods.”
I enjoyed reading and reviewing this paper which is thought provoking and helps to fill the gap in evidence-based knowledge to guide paediatricians and others involved in the counselling of parents and also those who advocate or set guidelines for adequate nutrition beyond 6 months in both breastmilk-fed and non-breastmilk-fed infants.
We are truly honoured by this reviewer's latest comment and, if the reviewer enjoyed reading and commenting on our work, we have read and accepted his comments with great pleasure.
Round 2
Reviewer 1 Report
The revised manuscript is improved. However, additional minor modifications can improve the manuscript and should be considered:
- Line 129: Replace “composition” with “content”
- Line 150: Replace the statement “The evaluated nutritional needs and recommendations are restricted to protein, calcium and iron for their major impact…” with “The evaluated nutritional needs and recommendations are specific to protein, calcium and iron regarding their major impact…”
- Line 196: Replace “by” with “in”
- Line 424: Replace “must” with “should”
- Line 428: Replace the statement “the nutritional analysis to "extreme" situations” with “the nutritional analysis to specific types of milk”
- Line 431: Replace “unnecessarily lengthened” with “excessively long”
- Line 438: Replace “best” with “better”
Author Response
Dear reviewer, thanks a lot for your help. We really appreciate it, and all the lines you pointed out have now been emended following your suggestions.
We thought it appropriate to thank all of you reviewers officially in the thanks section of the article for the positive comments you have all had for our work.
Best regards, the Authors
Reviewer 2 Report
Thank you for addressing some of my comments. This journal is an international journal and the information about formulas is available via the internet. It would therefore be possible to address my comments but I accept that you decline this.
Author Response
Dear reviewer, thanks a lot for your understanding of our difficulties. We really appreciate it. We are fully aware of the international scope of Nutrients, and sincerely believe we respected it, since formulas’ compositions we took into account are ruled (and in use) all around the European Union. Nonetheless, your suggestion may give hope for a cooperation in building up a comparative evaluation using a wider choice of formulas marketed in wider areas of the world. Should you been interested in such a cooperation, please be so kind as to contact us (since the reviews are blinded, we don’t know who you are, and we cannot be the first ones to contact you, sorry for that).
We thought it appropriate to thank all of you reviewers officially in the thanks section of the article for the positive comments you have all had for our work.
Best regards, the Authors